# High-resolution sampling of beam-driven plasma wakefields

S. Schröder [1,2 ✉], C. A. Lindstrøm [1], S. Bohlen [1,2], G. Boyle [1], R. D'Arcy[1], S. Diederichs [1,2], M. J. Garland[1], P. Gonzalez [1,2], A. Knetsch [1], V. Libov[1], P. Niknejadi [1], K. Pøder [1], L. Schaper [1], B. Schmidt[1], B. Sheeran[1,2], G. Tauscher[1,2], S. Wesch [1], J. Zemella[1], M. Zeng [1] & J. Osterhoff [1]

Plasma-wakefield accelerators driven by intense particle beams promise to significantly reduce the size of future high-energy facilities. Such applications require particle beams with a well-controlled energy spectrum, which necessitates detailed tailoring of the plasma wakefield. Precise measurements of the effective wakefield structure are therefore essential for optimising the acceleration process. Here we propose and demonstrate such a measurement technique that enables femtosecond-level (15 fs) sampling of longitudinal electric fields of order gigavolts-per-meter (0.8 GV m$^{-1}$). This method—based on energy collimation of the incoming bunch—made it possible to investigate the effect of beam and plasma parameters on the beam-loaded longitudinally integrated plasma wakefield, showing good agreement with particle-in-cell simulations. These results open the door to high-quality operation of future plasma accelerators through precise control of the acceleration process.

[1] Deutsches Elektronen-Synchrotron DESY, Notkestraße 85, 22607 Hamburg, Germany. [2] Universität Hamburg, Mittelweg 177, 20148 Hamburg, Germany. ✉email: sarah.schroeder@desy.de

Plasma wakefields[1] can accelerate charged particle beams with gradients in excess of gigavolts-per-meter, promising more compact accelerators for high energy physics and photon science[2,3]. In a beam-driven plasma-wakefield accelerator[4,5], a high-density charged-particle beam interacts with plasma electrons to form an electron density wake, in which trailing particles can rapidly gain energy from strong electric fields[6]. The resulting energy spectrum of the accelerated particles is determined by the detailed structure of this plasma wakefield, which again depends on the exact distributions of plasma density and beam charge[7]. Free-electron lasers[8] and particle colliders[9] have strict demands for precise control of the energy spectrum (e.g., low spread of energies), which have not yet been met by plasma accelerators. To reach this level of precision, it is first necessary to measure the wakefield with high resolution—highly nontrivial at the required plasma densities. Here we present a method that enables such a high-resolution sampling of the effective wakefield in a beam-driven plasma accelerator. The method is based on separating the energy and time measurements of the longitudinal phase space of particle beams by energy collimation of chirped bunches, thereby overcoming the practical challenges faced by previous methods of temporally resolving bunches after plasma interaction. Using this technique, we demonstrate experimentally how beam and plasma parameters affect the wakefield. The method enables a new level of precision optimisation of the plasma acceleration process necessary to attain the lower energy spread, higher efficiency[10], and higher transformer ratios[11,12] desired for future applications.

Various techniques exist for measuring the structure of a plasma wake[13]. The electron density distribution can be imaged using laser-based methods like shadowgraphy[14] and frequency domain holography[15]. Also the magnetic fields inside the wake can be measured using lasers via the Faraday effect[16,17]. However, a direct measurement of the electric fields requires the use of charged particles. This can be done by traversing the wake perpendicular to its direction of motion with a short probe electron bunch, such that the transversely-integrated wakefield is imprinted on the transverse profile of the probe[18]. Alternatively, in a beam-driven plasma accelerator the beam itself can be used to measure the longitudinal wakefield, by comparing the longitudinal phase space (i.e., a time-resolved energy spectrum) of bunches with and without plasma interaction. This gives access to the effective wakefield, longitudinally integrated over the full plasma accelerator module—ultimately the quantity that needs to be optimised. The longitudinal phase space is typically measured using a magnetic spectrometer in combination with either a streak camera or an RF transverse deflecting structure (TDS)[19,20]. Both methods have been used to measure wakefields[11,12,21] in plasmas with densities of order $10^{13}$–$10^{14}$ cm$^{-3}$, requiring picosecond resolution—close to the resolution limit of a streak camera, whereas a TDS can, in principle, provide time resolution down to the femtosecond-scale.

Reaching high gradients for compact acceleration requires operating at plasma densities of $10^{16}$–$10^{17}$ cm$^{-3}$. In this range, prior to being optimised, the plasma wakefield can generate large energy spreads and complex, slice-dependent distributions in transverse phase space—highly divergent beams that are difficult to transport. As a result, measuring the longitudinal phase space of the plasma-interacted bunches becomes very challenging. The energy spectrum is therefore commonly measured close to the plasma module to avoid excessive chromaticity. This makes conventional use of a TDS highly impractical: the required ultrahigh-vacuum conditions cannot easily be met in the vicinity of a gas load such as a high-density plasma cell; the structure can be damaged by irradiation; and the diverging beams can sample off-axis longitudinal RF fields[22]. Alternative methods for providing the required

time resolution are therefore needed. The first example of this was presented by Clayton et al.[23], who were able to use correlations within the complex energy transverse phase space measured on a spectrometer screen to measure the wakefield indirectly at high density. However, this particular method works only for mismatched beams (causing emittance growth[24]) and the temporal resolution is highly restricted.

In this paper, we propose and demonstrate a more general solution to measuring beam-driven plasma wakefields—by fully separating the time and energy measurements of the longitudinal phase space. The method works by linking the two measurements using an energy collimator and bunches with a highly correlated (chirped) longitudinal-phase-space distribution. This chirp allows slice-by-slice collimation of the incoming beam-current profile (see Fig. 1), which can be used to progressively remove thin tail slices from the energy spectrum after plasma-interaction. Full time-resolvability necessitates a detailed temporal calibration of the collimators prior to the measurement (see Methods) and is then achieved by comparing the spectra as the collimator position is varied to find the energy of the charge that disappears between steps. Crucially, the measurement of the longitudinal slice position within the bunch for each collimator step can now be entirely

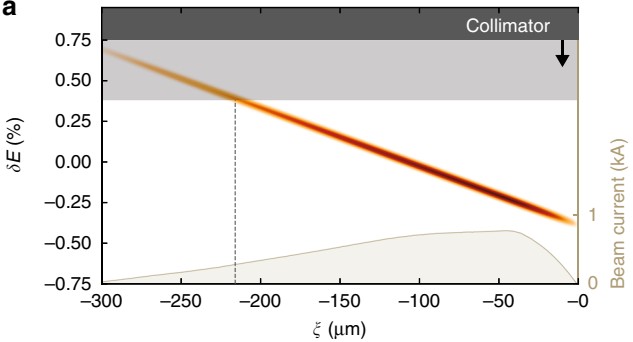

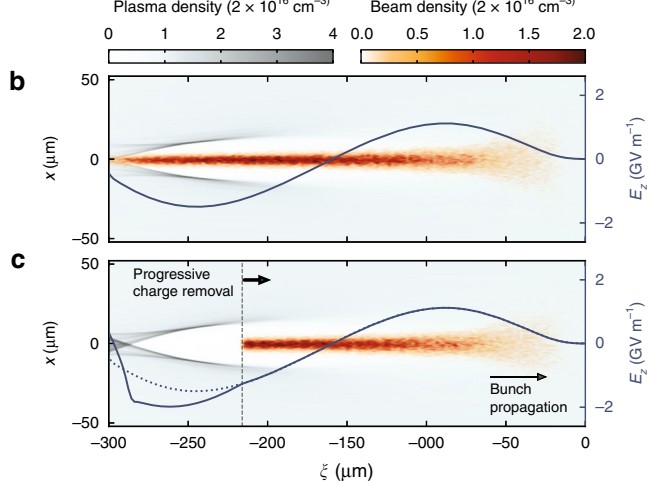

**Fig. 1 Plasma wakefield sampling by energy collimation. a** A strongly correlated longitudinal phase space allows tail slices to be progressively removed from the bunch, e.g., by collimation in a dispersive section. **b** An electron bunch interacts with a plasma and excites a density wake with strong longitudinal electric fields (3D particle-in-cell simulation). **c** Removing charge from the bunch tail alters the wakefield (compared to the original field, dotted line), but does not alter the wakefield experienced by the remaining bunch charge. By subtracting the final energy spectrum of consecutive collimator steps, the energy change of each longitudinal slice can be determined—revealing the effective wakefield.

disentangled from the energy-spectrum measurement. As a consequence, this temporal measurement (e.g., using a TDS) can be performed elsewhere in the beamline, where conditions such as beam optics and vacuum requirements can be optimised for maximum time resolution, and not for the operation of a plasma accelerator. The measurement can even be performed in a parallel beamline if the difference in the longitudinal phase space can be accurately determined.

While bunch-tail removal does affect the wakefield, it does not influence the part being sampled—the wakefield experienced by the removed tail slice is identical to that experienced within an uncollimated bunch (see Fig. 1c). This is because the beam and the plasma wake both travel at approximately the speed of light, and therefore (by causality) cannot be affected by changes behind the cut. By using energy collimation to manipulate the current profile, some constraints are imposed—in particular, a strictly monotonic longitudinal-phase-space distribution is required. In general, a linear monotonic correlation will result in an equidistant sampling of the wakefield for a constant collimator step size. Non-linear correlations are applicable, but sample non-equidistantly. Moreover, the accuracy of the measurement will be limited by the sliced energy spread of the chirped bunches relative to their overall energy spread, as well as the beam-size-to-dispersion ratio at the location of the collimator.

## Results

**Experimental setup.** This wakefield sampling measurement technique was implemented and experimentally demonstrated at the FLASHForward plasma accelerator facility[25] at DESY. High-quality electron bunches were generated with a photocathode at a repetition rate of 10 Hz and accelerated to 1.1 GeV using super-conducting RF cavities in the FLASH linac[26]. The bunches were linearised in longitudinal phase space and compressed to a bunch length of 285 ± 2 fs rms, characterised using an S-band TDS in a beam line parallel to the FLASHForward experimental area. The beam was then extracted into a dispersive section with a set of energy collimators for advanced shaping of the current profile[27]. The bunch head was collimated throughout the experiment to optimise the plasma interaction. A downstream toroid measured

the total delivered charge to be 460 ± 5 pC. Two sets of quadrupoles were used to focus the beam tightly at the interaction point, where the beam orbit was measured using two cavity-based beam-position monitors (BPMs). The plasma was generated by a high-voltage discharge in a 1.5-mm-diameter, 33-mm-long sapphire capillary filled with argon at a backing pressure of 40 mbar. The plasma cell was separated from the accelerator vacuum by three (windowless) differential pumping stations. After exiting the plasma cell, the energy spectrum of the electron beam was measured with a beam-imaging spectrometer, consisting of a vertically-dispersive dipole magnet and a set of quadrupoles for point-to-point beam imaging from the plasma exit plane to a scintillating LANEX screen. Figure 2 shows the experimental setup (see Methods for more details).

**Experimental campaign.** The wakefield sampling measurement consisted of a set of tail-collimator scans, progressively removing the rear part of the bunch until no charge was left. The plasma density was chosen such that the first oscillation of the plasma wake could be probed by the full length of the current profile (a density of $\sim 2 \times 10^{16}$ cm$^{-3}$). The evolution of the energy spectrum of the collimated bunches is shown in Fig. 3. The scan was repeated at three different imaging energies to ensure good energy resolution over the entire spectrum. The final energy of each longitudinal slice was then determined by the spectral difference between consecutive collimator steps (see Fig. 3c), resulting in a distinct signal that can be programmatically extracted (see Methods). The tail-collimator scan was then repeated with the plasma turned off, in order to accurately determine the energy change of each longitudinal slice. Some noise appears in the difference measurement due to imperfect stability of the plasma acceleration process, but this does not significantly hinder the extraction of the wakefield signal.

Time calibration of the tail-collimator steps was performed by direct comparison to the longitudinal phase space measured by the TDS. The cumulative charge below each energy in the longitudinal phase space was compared to the charge readings of the toroid downstream of the tail collimator (see Methods). Additionally, a known compression factor of 1.09 (i.e., 9% shorter

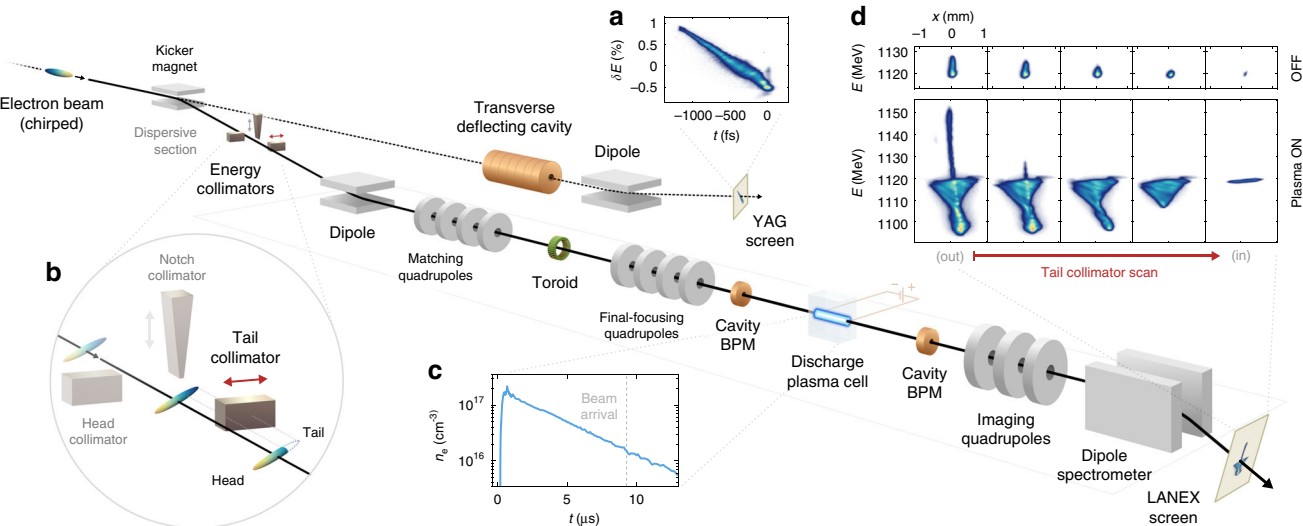

**Fig. 2 Experimental setup. a** An electron beam with a linearly chirped longitudinal phase space is characterised in a parallel beam line using a TDS. **b** A beam enters a dispersive section with energy collimators for optional removal of the tail (high energy), the head (low energy), and the centre (for two-bunch generation). The beam charge is measured with a toroid after collimation and before the beam is focused into a discharge plasma cell. **c** The relative time-of-arrival of the beam is adjusted to reach the desired plasma density: the temporal evolution of the average density was measured using two-colour interferometry in a test stand with a replica cell. **d** Finally, the beam following plasma interaction passes through a quadrupole triplet and is vertically dispersed by a spectrometer dipole to form a point-to-point image on a scintillating LANEX screen.

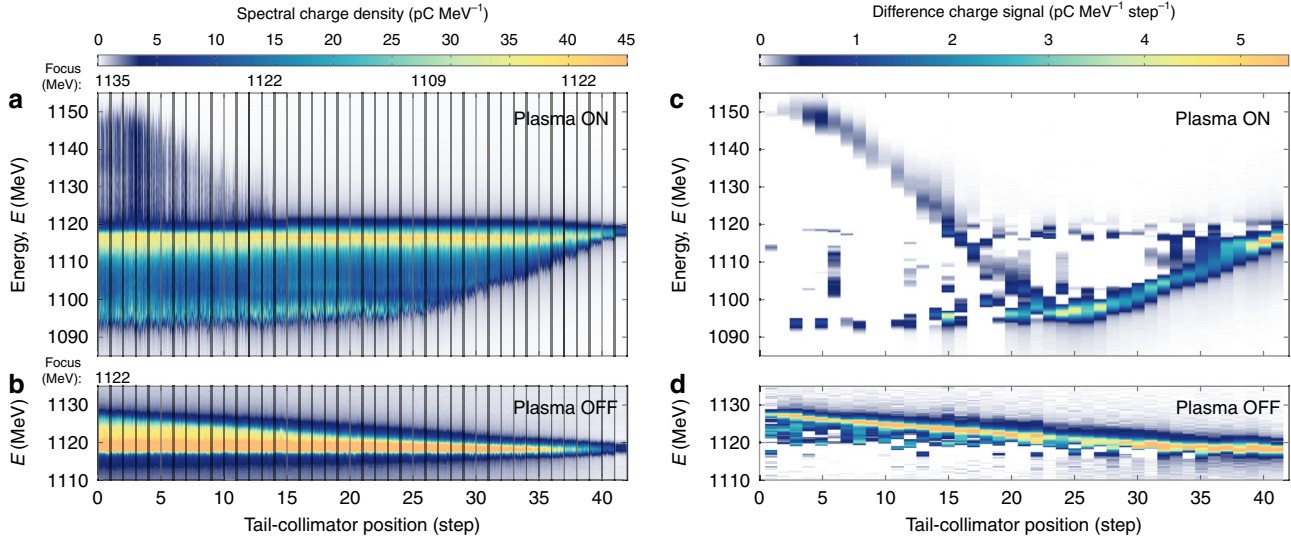

**Fig. 3 Evolution of the tail-collimated energy spectrum. a** As the tail collimator is moved in (horizontal axis), the tail particles are progressively removed from the plasma-interacted energy spectrum; 70 shots are collected per step. **b** The same scan is repeated with no plasma (20 shots per step), demonstrating the energy collimation of the incoming bunch. **c** The difference of (averaged) spectra at consecutive collimator steps reveals the energy of each slice of the plasma-interacted bunch. **d** By subtracting the plasma-off energy of each slice, the wakefield signal can be extracted.

bunches) from the calculated longitudinal dispersion $R_{56}$ of the extraction sections was taken into account.

**Interpretation of experimental data**. Figure 4a shows the resulting time-resolved wakefield measurement, sampled with more than 40 data points separated by ~6 μm. The electric field is calculated as the slice energy change normalised by the full 33-mm length of the cell—giving the effective wakefield of the plasma module. The wakefield is observed to have a zero-crossing at −165 μm (behind the bunch head), and reaches a maximum average accelerating gradient of $0.82 \pm 0.08$ GV m$^{-1}$ at the bunch tail.

Given that the effective wakefield is longitudinally averaged, understanding the full evolution of the beam and the wakefield requires a 3D particle-in-cell (PIC) simulation (see Methods). Since the measurement is precise to the few-percent level, a similarly detailed understanding of every aspect of the simulated system is required—in particular the 6D phase space of the beam, as well as the longitudinal density profile of the plasma. Using quadrupole scans and a two-BPM technique for determining the transverse phase space[28], sliced transverse beam parameters were estimated (see Methods). Additionally, a longitudinally resolved measurement of the plasma density was performed in a replica cell using Stark broadening[29], where the argon was doped with hydrogen (see Methods). Due to the Gaussian-like longitudinal plasma-density profile, the wakefield and, as such, the peak electric-field amplitude evolve along the length of the plasma channel. The simulations indicate that the largest instantaneous accelerating field observed was −1.8 GV m$^{-1}$ at a plasma density of $2.6 \times 10^{16}$ cm$^{-3}$, 7 mm after the peak of the Gaussian-like longitudinal density profile. This implies that the plasma accelerator was operating in a quasi-linear regime, reaching ~12% of the wave-breaking field. As a result of this detailed characterisation of the beam and plasma parameters, a good agreement between simulation and measurement was achieved (as seen in Fig. 4)—lending credibility to the accuracy of the method.

Having demonstrated the precise measurement of plasma wakefields, we can now investigate the effects of changing key parameters: (1) the plasma density, and (2) the current profile. Figure 4b shows the measured wakefield for a 80% higher plasma

density. The wavelength is observed to decrease (zero-crossing at −155 μm) and the wakefield amplitude increases, as expected. Figure 4c shows the effect of introducing a notch collimator (see Fig. 2b) to remove the central part of the bunch—producing a double-bunch current profile. This does not alter the wakefield ahead of the notch, but drastically changes the shape of the wakefield experienced by the trailing charge. In this case, the zero-crossing of the wakefield occurs much earlier (at around −130 μm) because the lack of beam current allows the expelled plasma electrons to start returning to the axis earlier (the wakefield is proportional to the radial velocity of the plasma-sheath electrons[7]). This latter measurement is a direct demonstration of the physics of beam loading[30], where the presence or absence of beam electrons alters the shape of the plasma wakefield.

## Discussion

The precise measurement of the plasma wakefield is an essential precursor to its optimisation. By using an energy-collimation technique, we have demonstrated that the necessary precision can be achieved—overcoming the practical challenges faced in a beam-driven plasma accelerator operating at high densities. Combined with an ability to fine-tune beam and plasma parameters, this innovative method opens the door to achieving milestones that require precisely tailored wakefields—such as optimised beam loading[31] for energy-spread preservation and high efficiency—while operating at the plasma densities and accelerating gradients necessary for many applications in high energy physics and photon science.

## Methods

**Generation of electron bunches**. The superconducting linear accelerator FLASH was used to accelerate electron bunches of 600 pC total charge to a mean particle energy of 1122 MeV. The bunches were compressed in two magnetic bunch-compressor chicanes to a peak current of 750 A. The longitudinal phase space of the bunch was measured by a transverse deflecting structure in a beam line parallel to the FLASHForward experimental area. The measured energy spread spans ~1%, with slice energy spreads well below 0.1% (the measurement is limited by the transverse beam size inside the TDS and on the screen). A kicker magnet was used to extract the beam into a dispersive section where a set of three collimators[27] were used for energy profile manipulation, which due to the strongly correlated longitudinal phase space also allows precise manipulation of the bunch current profile.

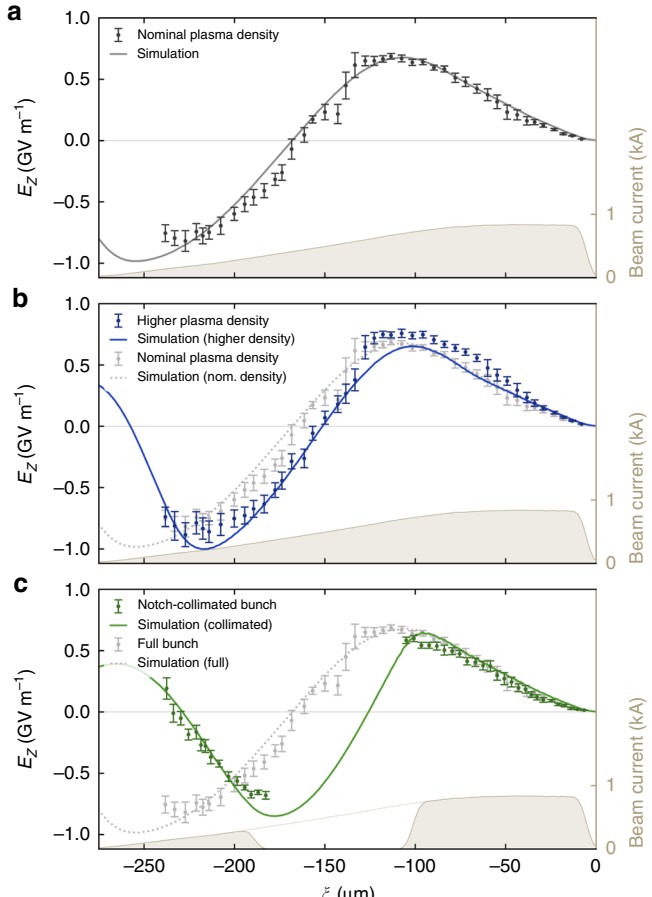

**Fig. 4 Measured longitudinally averaged plasma wakefields. a** The wakefield of the nominal beam and plasma is sampled with a 6 μm granularity. Error bars represent the shot-to-shot standard deviation. The solid line shows the corresponding PIC simulated wakefield. **b** The measurement is repeated with the same beam current profile and an 80% higher plasma density (the beam arrives 1.7 μs earlier), producing a shorter wavelength wakefield. **c** At the nominal density (same as in **a**), the beam current is notch-collimated into a double-bunch profile, producing a wakefield which is identical at the head, but strongly altered at the tail. The end point of $\xi = -240$ μm is determined by signal-to-noise ratio.

Owing to collective effects during bunch compression, it was found that collimating the bunch head (down to a total charge of 460 pC) increases the stability of beam–plasma interaction. This setting was chosen prior to the measurement campaign and kept throughout it. Toroids were used to measure the beam charge before and after the energy collimation. The bunches were further compressed by an estimated 9% in the dispersive extraction line. A set of nine quadrupoles was used to tightly focus the beam at the location of the plasma cell. Two cavity-based beam-position monitors (50 cm upstream and 50 cm downstream of the plasma) were used for beam alignment. Three differential pumping stations enabled a windowless vacuum-to-plasma transition—ensuring high beam quality while also meeting the ultrahigh vacuum requirements of the superconducting FLASH accelerator.

**Plasma source.** A discharge plasma source was used to create the plasma, ignited by a high-voltage thyratron supplying ~500 A of current at 25 kV for a duration of 400 ns. A thin 1.5-mm-diameter, 33-mm-long capillary was milled from two slabs of sapphire, mounted in a PEEK plastic holder, again mounted on a hexapod platform for high-precision alignment. A continuous flow of argon was supplied through two internal gas inlets from a buffer volume at a 40 mbar backing pressure. The gas escaped the open-ended capillary through holed copper electrodes (cathode upstream, anode downstream) into a large 500-mm-diameter vacuum chamber pumped to an ambient pressure of $4.3 \times 10^{-3}$ mbar.

**Electron imaging spectrometer.** A dipole magnet was used to perform energy dispersion of the beam vertically onto a LANEX (fine) screen mounted just outside

the 1-mm-thick stainless-steel vacuum chamber wall, ~3 m downstream of the plasma cell. Five quadrupoles (acting as a triplet) located just upstream of the dipole were used to point-to-point image the beam from the plasma cell exit (the object plane) to the screen (the image plane) with a magnification of $R_{11} = -5$ (horizontally) and $R_{33} = -0.43$ (vertically), where $R$ is the object-to-image-plane transfer matrix. The spatial resolution of the optical system was ~50 μm (i.e., ~2 pixels), corresponding to an energy resolution of 0.05% for particles close to the imaged energy. Away from this imaged energy, the energy resolution degrades depending on the vertical divergence of the bunch.

**Wakefield signal extraction.** The wakefield signal of a bunch slice is given by the difference in the energy spectrum between two consecutive collimator steps. All energy spectra (shots $i$) at collimator step $s + 1$ are subtracted from all energy spectra (shots $j$) at collimator step $s$, and subsequently fitted with a Gaussian distribution. Difference signals with a Gaussian peak $\mu_{ij}$ further away than 10 MeV from an initial peak estimate are rejected. Each event $i$ of collimator step $s$ has an attributed mean signal at $E_i = \frac{1}{n}\sum_{j=1}^{n}\mu_{ij}$, where $n$ is the number of shots per step. The resulting energy signal between collimator steps $s$ and $s + 1$ is given by the mean energy $E_{s+1/2} = \frac{1}{n}\sum_{i=1}^{n} E_i$. The standard deviation of $E_s$ is used as the uncertainty of the signal. This uncertainty represents statistical fluctuations and is connected to the stability of the beam–plasma interaction. The same wakefield-signal extraction routine is used for all datasets. The presented wakefields are spliced from three datasets imaging at low, middle and high energies onto the spectrometer screen (1109, 1122 and 1135 MeV, respectively).

**Temporal calibration of collimator steps.** The collimator position scan is mimicked by a virtual energy-collimation scan on the reconstructed longitudinal-phase-space measurement (Supplementary Fig. 1a). The subtraction signal of two current profiles of consecutive collimator positions is fitted with a Gaussian distribution, where the peak position is used for the collimator-to-longitudinal-position calibration. Comparing the remaining charge measured in both the real and the virtual collimator scans (Supplementary Fig. 1b) can then be used to determine the corresponding longitudinal position within the bunch at each individual collimator step (Supplementary Fig. 1c). An additional compression factor of 1.09 from the extraction line is also taken into account—this factor is determined by the lattice configuration ($R_{56} = -3$ mm) in the dispersive section of the beamline.

**Transverse beam characteristics.** The transverse phase space of the beam was measured using a two-BPM tomography technique[28]. The phase space of the centroid jitter measured with two BPMs (upstream and downstream of the plasma chamber) allows the Twiss parameters of the beam to be estimated—a measurement based on the observation that the Twiss parameters of the jitter phase space are similar to those of the beam phase space. Combining this measurement with a head-and-tail-collimator position scan, which allows passage of 0.1% rms energy slices, enables an energetically resolved (and because of the strong chirp also temporally resolved) characterisation of the transverse phase space. The sliced emittance in the horizontal plane was determined using a quadrupole scan on the electron spectrometer, indicating a normalised emittance between 1 mm mrad (tail) and 10 mm mrad (head)—a variation likely to be caused by transverse kicks from coherent synchrotron radiation (CSR). The emittance in the vertical plane could not be measured directly, but was estimated to be 0.5–1 mm mrad—a typical value measured directly at the gun, not expected to be affected by CSR during transport. The beam was focused to beta functions of ~10 × 10 mm², with a waist close to the plasma entrance, with a relatively large chromaticity (correlation between slice energy and waist location) in the vertical plane caused by asymmetric focusing in the final-focusing quadrupoles.

**Longitudinal plasma density profile.** Two complimentary diagnostics techniques were used to characterise the density profile evolution of the argon plasma[29]. Two-colour laser interferometry was used to measure the longitudinally integrated average plasma density (see Fig. 2c), and Stark broadening allowed the density to be longitudinally resolved. This was performed with a replica cell in a dedicated test laboratory. Supplementary Figure 2 shows the profile and evolution of the plasma density between 4 and 10 μs after the discharge, where the argon was doped with 5% hydrogen to enable Stark broadening of the H-alpha line. The density measurements were fitted (see Supplementary Fig. 2a) to obtain longitudinal profiles at 7.5 and 9.3 μs, which were then scaled to the corresponding absolute average density in pure argon (shown in Supplementary Fig. 2b).

**Particle-in-cell simulations.** The 3D quasistatic particle-in-cell code HiPACE[32] was used to simulate the full evolution of the beam–plasma interaction. The input beam was generated based on the 6D-phase-space information of the experimentally characterised beam. It was modeled with $0.2 \times 10^{6}$ constant-weight numerical particles. Similarly, a 33-mm-long longitudinally tailored plasma-density profile was implemented based on density measurements (see above). The plasma was sampled with 4 particles per cell. A simulation box of size $20 \times 20 \times 7$ $k_p^{-3}$ (in $x \times y \times \xi$) was resolved by a grid of $1024 \times 1024 \times 200$ cells, evolved with a

constant time step of 3.5 $\omega_p^{-1}$, where $k_p$ and $\omega_p$ are the plasma wavenumber and frequency, respectively. The resulting longitudinally averaged, on-axis electric fields were used for comparison to the experimentally measured wakefields (see Fig. 4).

## Data availability
The data that support the findings of this study are available from the corresponding author upon reasonable request.

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

## Acknowledgements
We gratefully acknowledge the Gauss Centre for Supercomputing e.V. (www.gauss-centre.eu) for funding this project by providing computing time through the John von Neumann Institute for Computing (NIC) on the GCS Supercomputer JUWELS at Jülich Supercomputing Centre (JSC). This work was supported by the Helmholtz ARD program. The authors would like to show their gratitude to the FLASH Leadership as well as the DESY FH and M divisions for their scientific, engineering and technical support.

## Author contributions
S.S. and J.O. conceived the idea of the wakefield sampling method. S.S. and C.A.L. carried out the experiments with the assistance of S.W., J.Z, P.G. and V.L., S.S. performed the data analysis with the assistance of J.O. and C.A.L., C.A.L. determined the transverse beam characteristics. S.S. performed the simulations with assistance of S.D. and M.Z. L.S., G.T. and M.J.G. set up the plasma density diagnostics with the assistance of S.B. and K.P. M.J.G. measured the longitudinal plasma density profile. G.B. developed the model for the plasma density extrapolation. S.S. and C.A.L. prepared the paper with the assistance of J.O., R.D., B.Schmidt and G.B.. C.A.L. created the equipment diagram Fig. 2. S.W., V.L., S.S., J.Z., R.D., P.N., L.S., A.K and B.Sheeran contributed to the commissioning of the beam line. J.O., R.D. and B.Schmidt supervised the project. All authors discussed the results presented in the paper.

## Funding

## Competing interests
The authors declare no competing interests.
