## [Peer Review File · Nature Communications]

REVIEWER COMMENTS

Reviewer #1 (Remarks to the Author):

The manuscript written by Schroeder et al presents a new method for mapping the beam driven plasma wakefield structure by a combination of an energy collimator and an electron bunch with highly-correlated longitudinal phase space distribution. The experiment results conducted at FLASHForward have shown that this method could offer a high-resolution sampling of the longitudinal electric fields in the plasma. The manuscript is well-written and logically organised. The data analysis and interpretation are reasonable.

Sampling of the plasma wakefield structures is crucial for any future applications (e.g. for light sources, colliders) of the plasma-based accelerators. This allows operators to optimize the beam and plasma parameters so as to achieve the specific and reproducible beams for the end users. There have been quite a few methods developed in the community to mapping the wakefield structures, such as frequency domain holography, shadowgraphy, electro-optic method or even using a short probe electron bunch (with femtosecond resolution in the plasma density of 10^{16} - 10^{17} cm⁻³), as some of them were pointed out by the authors in the manuscript.

The new method proposed and demonstrated in this manuscript proved its effectiveness. However, there are some limitation (drawbacks) for its wide applications. Firstly, this method requires a strictly monotonic (strongly correlated) longitudinal phase space distribution. However, the conventional accelerators in most cases cannot provide such ideal beams due to many reasons. What will happen if the output beam is an S-shaped? The authors have not elucidated in the manuscript. Secondly the proposed method uses the collimator to remove part of particles in the bunch head and tail to optimize the plasma interaction. This will somehow reduce the wakefield due to less particles contributing to the interactions with the plasmas.

In addition, the visibility of Figure 4 can be improved, especially some lines are very weak to be seen. In lines 230-232, the authors claim that the field amplitude (accelerating field) is as high as -1.8 GV/m. However Figure 4 does not shows such high field, why?

This manuscript contains the new method and results; however, the reviewer think it is not enough to be published to Nature Communications as its novelty is not so high. I would suggest that authors publish this research results to Scientific Reports.

Reviewer #2 (Remarks to the Author):

This paper addresses one of the most important questions facing the modern-day plasma wakefield accelerators - acceleration of the electron beams to controlled energy and with small energy spread. Unrepetable energy gains and very large energy spreads in electron means emerging from the plasma accelerator is the main weak point of plasma wakefield accelerators, when one compared

them with rock-stable conventional accelerators producing beams with energy spreads down to 0.01%. As authors properly pointed out, accurate measurements of the wakefields are absolutely essential to pinpoint weakness of existing accelerating schemes and to find way for improving the beam quality from plasma accelerators.

The paper is very well written, all necessary details are of rather ingenious methods are explained well and then accurately implemented in the experimental set-up. Results are both excellent and important for the field. The fact that simulations agree reasonably well with the experimental results is very nice - it can be stated that the simulation code is now properly vetted.

I also was impressed by authors attention to details and well written Methods section.

To conclude, I consider this to be excellent paper that must be published in this journal.

I am sure that impact of this paper o the field of plasma accelerators will be dramatic!

Reply to the Reviewers

We would like to thank the referees for their efforts to review the manuscript and very much appreciate their highly valuable comments. We are grateful for their support and acknowledgement of the importance of an accurate wakefield measurement and we reply in detail below to their criticism.

Note: in the following the Reviewers' comments are marked in green on grey background, followed by our replies in black, with corresponding changes to the manuscript in bold. The stated line numbers refer to the revised version of the manuscript, in which changes are highlighted in blue.

Authors' reply to **Reviewer #1**

Reviewer #1 overview

„The manuscript written by Schroeder et al presents a new method for mapping the beam driven plasma wakefield structure by a combination of an energy collimator and an electron bunch with highly-correlated longitudinal phase space distribution. The experiment results conducted at FLASHForward have shown that this method could offer a high-resolution sampling of the longitudinal electric fields in the plasma. The manuscript is well-written and logically organised. The data analysis and interpretation are reasonable.

Sampling of the plasma wakefield structures is crucial for any future applications (e.g. for light sources, colliders) of the plasma-based accelerators. This allows operators to optimize the beam and plasma parameters so as to achieve the specific and reproducible beams for the end users. There have been quite a few methods developed in the community to mapping the wakefield structures, such as frequency domain holography, shadowgraphy, electro-optic method or even using a short probe electron bunch (with femtosecond resolution in the plasma density of 10^{16} - 10^{17} cm⁻³), as some of them were pointed out by the authors in the manuscript.“

A number of methods exist to visualise and diagnose different aspects of the structure of a plasma wake. As pointed out by the Reviewer, high-resolution sampling of the *electric field* in plasma, as experienced by the accelerated beam, plays a pivotal role for the future applicability and impact of plasma-based accelerators. Only when these fields are precisely measured, can they be controlled with high-precision to achieve reproducible beams with tuned energy distributions. Crucially such measurements need to diagnose the *beam-loaded field at the location of the accelerated beam inside the wake integrated over its acceleration path*. This is exactly what the method described in this study enables us to do. Using this new method, the effect of beam and plasma parameters on a wakefield with GV/m amplitude was directly investigated for the first time.

All of the other mentioned techniques have their use cases and can reveal various details of strong plasma wakes but none of them simultaneously provide the above discussed capabilities and deliver the potential for direct feedback for optimisation of the beam energy distribution. Frequency domain holography [15] and shadowgraphy [14] have been used to detect *gradients in the plasma refractive index* and, thus, can reveal *the electron density distribution*, Faraday-rotation techniques [16, 17] were deployed to probe *transversely integrated snapshots of the magnetic field distribution*, and short probe electron bunches [18] were utilised to sample the *transversely-integrated wakefield*. *Longitudinal electric field strength information* was previously extracted from correlations in the complex transverse phase space of beams which are not transversely matched to the wakefield [23]. This latter scheme provides limited temporal resolution and, in contrast to our new sampling method, is not applicable under conditions which support the preservation of transverse beam emittance—which practically precludes its usage for all relevant applications in photon science and particle colliders.

We therefore believe that our method bridges the considerable gap in diagnostic capabilities to allow for accurately measured and, consequently, tailored beam-loaded wakefields with GV/m amplitude. As such, this approach is novel and will enable plasma-wakefield accelerators to enter the realm of high resolution, precision tuning with great impact on their applicability.

In order to more clearly highlight the aspect of novelty and to put our work into the discussed scientific context, the manuscript has been modified in lines 1ff and 45ff.

Reviewer #1 comment / limitations of method

„The new method proposed and demonstrated in this manuscript proved its effectiveness. However, there are some limitation (drawbacks) for its wide applications. Firstly, this method requires a strictly monotonic (strongly correlated) longitudinal phase space distribution.“

The Reviewer is correct that the longitudinal phase space (LPS) of the input electron beam requires a monotonic dependence of time and energy. High-energy electron injectors with 100-femtosecond-level bunch duration—parameters that are needed for high-gradient acceleration in plasma densities on the order of 10^{17} cm^{-3} —typically rely on bunch compression schemes that anyhow require LPS shaping. As such, most plasma-wakefield accelerator facilities (e.g. FACET-II, AWA, FLASHForward) have the intrinsic capability to shape the LPS such that the proposed diagnostic method could be implemented with minor modifications to the hardware setups and beam setup procedures. Furthermore, future high-energy facility based on beam-driven plasma-acceleration technology will most likely require such compression schemes with a dispersive section too—enabling collimation a simple and low-cost addition. Therefore, we do not believe the monotonic LPS to result in a significant limitation for the applicability of the scheme in terms of hardware compatibility.

Reviewer #1 comment / limitation of method — special LPS shapes

„However, the conventional accelerators in most cases cannot provide such ideal beams due to many reasons. What will happen if the output beam is an S-shaped? The authors have not elucidated in the manuscript.“

A monotonically correlated but non-linear phase shape results—for equidistant collimation steps—in a non-linear sampling of the wakefield. In general this does not harm the measurement. The calibration of the collimator step position to the corresponding zeta position (see Methods section) can be carried out also in this case with various tools such as an EOS, streak camera, or TDS.

For confirmation, simulations have been performed (see attached graph below): Beams with identical current profiles but different LPS shapes (left column) drive a wakefield. Consecutive removal of the bunch tail by energy collimation reveals the wakefield (middle column). An s-shaped LPS simply results in non-equidistant sampling positions (right column, bottom plot). In principle, these sampling positions could be made equidistant also for an s-shaped LPS by dynamically adapting the collimation step size—the achievable sampling granularity is then limited by the slice energy spread.

The necessity of a careful zeta-calibrations is further emphasised in line 108ff.

The case of a non-linear LPS shape is now elucidated in line 134ff.

Reviewer #1 comment / Method — change of wakefield due to bunch modifications

„Secondly the proposed method uses the collimator to remove part of particles in the bunch head and tail to optimize the plasma interaction. This will somehow reduce the wakefield due to less particles contributing to the interactions with the plasmas.“

Removing the charge from the tail does not alter the wakefield in front of the point of charge removal (as shown by the electric field lines in Figure 1). The particle beam is propagating practically at the speed of light such that changes at the bunch tail cannot propagate to and influence any part of the beam in front of it. By using a subtractive data analysis technique, the removed charge slice is hence probing the unaltered wakefield.

In general, removing the head of the bunch is not part of this method and is not required for it to function. In our particular case, the same part of the bunch head was removed in all the presented data sets to stabilize the beam-plasma interaction. The reduced stability of the unmodified beam (when the beam head was not removed) was most likely caused by collective synchrotron radiation effects arising from bunch compression inducing asymmetries in the transverse charge distribution. Removing the affected parts at the front of the beam prior to plasma interaction can mitigate this instability. This issue is explicitly not related to the wakefield measurement method itself. The amount of charge removed from the front of the beam was the same for all data sets presented here.

Changes have been made in lines 126, 157ff and 300ff to improve clarity.

Reviewer #1 comment / figure visibility

„In addition, the visibility of Figure 4 can be improved, especially some lines are very weak to be seen.“

Thank you for pointing this out, Fig. 4 has been improved.

Reviewer #1 comment / data discussion — field amplitudes

„In lines 230-232, the authors claim that the field amplitude (accelerating field) is as high as -1.8 GV/m. However Figure 4 does not shows such high field, why?“

Figure 4 shows the measured *longitudinally averaged wakefield*. Its average peak amplitude over the plasma cell length of 33 mm was measured to be ~ 0.8 GV/m. The Gaussian-like shape of the plasma density distribution (see Figure 6) over the plasma cell length results in varying *local field amplitudes* along the acceleration path. The maximum *local field amplitude* can be inferred from simulations to be -1.8 GV/m.

We have further clarified this by implementing changes to the manuscript in lines 216 and 232ff.

Reviewer #1 conclusion

„This manuscript contains the new method and results; however, the reviewer think it is not enough to be published to Nature Communications as its novelty is not so high. I would suggest that authors publish this research results to Scientific Reports.“

We earnestly believe that our new diagnostic method and the presented results constitute a significant development within the field, worthy of the impact level of this journal—a notion that is shared by Reviewer #2. We therefore thank Reviewer #1 for raising these criticisms as we believe our responses to them have served to better illustrate the novelty of the method we devised and have thus led to an increased extraction of impact from the results. Through these modifications and the increased level of clarity we hope that any fears of suitability stemming from a perceived lack of novelty are removed. As such, we hope that, upon further consideration, these changes and explanations will support the Reviewers' previous positive statements about the “new method and results” described here, ultimately resulting in a positive reconsideration on suitability for publication in Nature Communications.

Authors' reply to **Reviewer #2**

Reviewer #2 overview

„This paper addresses one of the most important questions facing the modern-day plasma wakefield accelerators - acceleration of the electron beams to controlled energy and with small energy spread. Unrepeatable energy gains and very large energy spreads in electron means emerging from the plasma accelerator is the main weak point of plasma wakefield accelerators, when one compared them with rock-stable conventional accelerators producing beams with energy spreads down to 0.01%. As authors properly pointed out, accurate measurements of the wakefields are absolutely essential to pinpoint weakness of existing accelerating schemes and to find way for improving the beam quality from plasma accelerators.

The paper is very well written, all necessary details are of rather ingenious methods are explained well and then accurately implemented in the experimental set-up. Results are both excellent and important for the field. The fact that simulations agree reasonably well with the experimental results is very nice - it can be stated that the simulation code is now properly vetted.

I also was impressed by authors attention to details and well written Methods section.

To conclude, I consider this to be excellent paper that must be published in this journal.

I am sure that impact of this paper o the field of plasma accelerators will be dramatic!“

We thank the Reviewer for this highly motivating and commending assessment of our work. We truly appreciate their reflection of the quality of the manuscript and of its future impact, as well as the support for our ambition of turning plasma accelerators into precisely controllable and applicable devices. Their statement that “*accurate measurements of the wakefields are absolutely essential to pinpoint weakness of existing accelerating schemes and to find way for improving the beam quality from plasma accelerators*” is of particular relevance since it highlights exactly how we believe the method presented in this manuscript will impact the field of plasma accelerators and its applications in the future.

REVIEWERS' COMMENTS

Reviewer #1 (Remarks to the Author):

Thanks to the authors have improved the manuscript based on our suggestion and comments. Various methods which have being used for diagnosing the structure of plasma wakefields are clearly explained and compared in the updated manuscript. In doing so, the advantage of the current method (i.e. using an energy collimator and an electron bunch with highly-correlated longitudinal phase space distribution) can be revealed. In addition, the authors have made it clear that this method can also be applied to other shapes of beam longitudinal phase space, for example, S-shaped. The authors clarified the bunch charge loss and its impact to the wakefield (due to collimation of bunch tail), the maximum field amplitude issues and improved the figure 4 accordingly. Through a careful reading, the current version of manuscript is much better.